# Modification of High-Density Polyethylene with a Fibrillar–Porous Structure by Biocompatible Polyvinyl Alcohol via Environmental Crazing

**DOI:** 10.3390/polym16091184

**Published:** 2024-04-23

**Authors:** Alena Yarysheva, Olga Arzhakova

**Affiliations:** Faculty of Chemistry, Lomonosov Moscow State University, Leninskie Gory 1-3, Moscow 119991, Russia; arzhakovaov@my.msu.ru

**Keywords:** environmental crazing, fibrillar–porous structure, polyvinyl alcohol, polyethylene, nanocomposites, biocompatible polymer

## Abstract

Polymer/polymer nanocomposites based on high-density polyethylene (HDPE) and biocompatible polyvinyl alcohol (PVA) were prepared by tensile drawing of HDPE in the PVA solutions via environmental crazing. The mechanism of this phenomenon was described. The HDPE/PVA nanocomposites were studied by the methods of scanning electron microscopy, atomic force microscopy, gravimetry, tensile tests, and their composition, properties, and performance were characterized. The content of PVA in the HDPE/PVA nanocomposites (up to 22 wt.%) was controlled by the tensile strain of HDPE and concentration of PVA in the solution. Depending on the content of PVA, the wettability of the HDPE/PVA nanocomposite (hydrophilic-lipophilic balance) could be varied in a broad interval from 45 to 98°. The modification of HDPE by the biocompatible PVA offers a beneficial avenue for practical applications of the HDPE/PVA composites as biomedical materials, packaging and protective materials, modern textile articles, breathable materials, membranes and sorbents, etc.

## 1. Introduction

Key properties and performance of polymers (biocompatibility, biodegradation, wettability, gas separation properties, antibacterial activity, mechanical performance, crystallizability) are controlled by their hydrophilic–lipophilic balance (HLB). The efficient approach providing an efficient hydrophilization of hydrophobic polymers is concerned with the preparation of polymer–polymer systems with polyvinyl alcohol (PVA) as blends, interpenetrating networks, and composites [1,2,3,4]. The high biocompatibility and hydrophilicity of PVA allows its application as biomedical materials, sorbents, gas separation and ultrafiltration membranes, packaging materials, polarizers, etc. [5,6].

However, the preparation of sustainable multicomponent polymeric materials presents a challenging scientific and technological task, especially when thermodynamically incompatible components with different physicochemical characteristics (hydrophilicity/hydrophobicity, melting temperature, the Hildebrand solubility parameter, etc.) are involved. Polymer–polymer blends can be prepared by different methods from the blending of polymeric solutions or melts and coextrusion [7,8,9] up to the development of interpenetrating networks via one-stage or two-stage polymerization [10,11]. It is noteworthy that all methods have their benefits and limitations, for example, the blending of PVA and polyolefins requires the use of plasticizing agents which markedly reduce the melting temperature of PVA [3,12,13]. The phase separation and aggregation of the components can be prevented by the addition of compatibilizers (maleic anhydride) [3,14].

This work addresses the preparation of HDPE/PVA nanocomposite materials via environmental crazing and highlights the benefits of this approach. The tensile drawing of amorphous glassy and semicrystalline polymers in the physically active liquid environments (PALE) proceeds via environmental crazing, which provides the development of a unique fibrillar–porous structure [15,16,17]. Environmental crazing is accompanied by a continuous stress-induced penetration of the PALE into the as-forming porous structure. When low-molecular-weight or high-molecular-weight compounds are dissolved in the crazing-promoting PALE, they are able to penetrate the fibrillar–porous structure, thus providing the development of a nanocomposite system [18,19,20].

In contrast to the above methods and approaches, environmental crazing allows the development of the mesoporous polymer matrix in one stage at room temperatures and the preparation of the nanocomposite via the introduction of PVA. In this case, the preparation of the multicomponent system containing a hydrophobic (HDPE) and a hydrophilic polymer (PVA) does not require the use of any compatibilizers, special reagents, or complex equipment (for example, extruders). The tensile drawing of the HDPE films can be performed using typical industrial equipment for the orientational drawing of polymers (with its minor modification).

The wettability of all multicomponent composite materials is known to be controlled by the HLB of the components. Hence, the introduction of the hydrophilic PVA to the hydrophobic HDPE matrix can result in the hydrophilization of HDPE. The problem of hydrophilization of high-tonnage commercial polymers presents a challenging task for modern materials and polymer science, and the solution of this problem makes it possible to broaden the scope of the practical application of HDPE and can also be considered as the approach for its modification and processing.

Thus, the objective of this work is concerned with the preparation of polymer–polymer composites based on hydrophobic and biocompatible hydrophilic polymers by the tensile drawing of HDPE films in PVA solutions and the characterization of the new HDPE/PVA materials (structure, wettability, vapor permeability, thermophysical characteristics, swelling, etc.).

## 2. Materials and Methods

### 2.1. Materials

In this work, we used commercial HDPE films (PBC “Sibur Holding”, Tobolsk, Russia; thickness of 60 µm; molecular mass 127 kDa; degree of crystallinity 63%) and PVA (EuroChem Group AG, Nevinnomyssk, Russia; molecular mass 150 kDa; trade mark 16/1). The dynamic viscosity of a 4% solution of PVA was 12–17 Pa·s·10^3^; the degree of saponification was 98.5%.

### 2.2. Preparation of HDPE/PVA Nanocomposite

The HDPE films with a gage size of 20 × 50 mm were stretched in the water–ethanol (3:2 = vol/vol) solution of PVA (the concentrations of PVA were 3, 7, and 11 wt.%) at room temperature; the strain rate was 5 mm/min. After stretching, the solvent was removed from the samples under isometric conditions by evaporation in vacuum for 3 days.

### 2.3. Methods

#### 2.3.1. Volume Porosity

To study the mechanism of plastic deformation of HDPE, the volume porosity (as the fractional content of pores) was estimated as:W = [ΔV/Vt] × 100 (%)(1)
where ΔV is the volume gain, and V_t_ is the current volume of the sample at a given tensile strain ε. The volume of the samples was estimated from the geometrical dimensions of the test samples (length, thickness, width) using an IZV2 optimeter; the experimental error was below ±0.5 μm (thickness). The measurements were performed for at least, 5–7 samples, and the results were averaged. The mean square error was 4%. The theoretical curve of an “ideal” crazing (under the assumption that plastic deformation proceeds via 100% crazing) was calculated as follows:W = [ε/(1 + ε)] × 100 (%)(2)

#### 2.3.2. Gravimetric Measurements

The content of PVA in the HDPE host matrix was calculated as:Δm/m_t_ = (m_t_ − m_0_)/m_t_ × 100 (%)(3)
where m_t_ is the weight of the HDPE/PVA nanocomposite, and m_0_ is the weight of the initial HDPE film. The measurements were performed for at least 5 samples, and the results were averaged. The mean square error was below 3%.

#### 2.3.3. Tensile Tests

Tensile tests were performed under uniaxial stretching of the HDPE and HDPE/PVA films at a constant strain rate of 5 mm/min using an Instron 4301 tensile machine (Instron Limited, Wycombe, UK). The samples were cut as dumbbell-shaped specimens with a gage size of 20 × 6 mm. The tensile drawing of the films in the PALE was performed using the sealed plastic bags. The measurements were performed for 5 samples, and the results were averaged. The mean square error was 10%.

#### 2.3.4. Strain Recovery

The strain recovery λ of the films after their tensile drawing via environmental crazing and stress relaxation when the samples were released from the clamps (free-standing state) was calculated as:λ = (L − L_t_)/(L − L_0_) × 100 (%)(4)
where L is the length of the sample after the tensile drawing, L_t_ is the length of the sample after stress relaxation in the free-standing state, and L_0_ is the initial length of the sample.

#### 2.3.5. Differential Scanning Calorimetry (DSC)

The thermophysical properties of the HDPE/PVA nanocomposites were studied with a TA400 thermal analyzer (Mettler-Toledo International Inc., Greifensee, Switzerland). The heating rate was 10 K/min. The test samples were cut from the central part of the films. The weight of the samples was 1–2 mg. The degree of crystallinity was calculated as:χ = ΔH/ΔH_χ=100%_ ×100 (%)(5)
where ΔH is the experimental heat of fusion, and ΔH_χ=100%_ is the heat of fusion of an ideal crystal (293 J/g for HDPE and 138.6 J/g for PVA).

#### 2.3.6. Atomic Force Microscopy (AFM)

The structure of the HDPE films was examined by the AFM method using a SolverPRO-M microscope (Nanotechnology MDT, Zelenograd, Russia) at the contact. The images were processed with the FemtoScan Online program (Center for Advanced Technologies, Moscow, Russia).

#### 2.3.7. Scanning Electron Microscopy with Energy Dispersive X-ray Spectroscopy (EDX SEM)

The distribution of PVA in the HDPE/PVA films was studied using a Hitachi S-520 scanning electron microscope (Tokyo, Japan). Prior to the SEM observations, the samples were fractured in liquid nitrogen. The fractured surfaces were decorated with gold using an Eiko IB-3 unit by ionic plasma deposition.

#### 2.3.8. Wettability

The hydrophilic–lipophilic balance (HLB) of the HDPE/PVA samples was characterized by the contact angle (CA) using a sessile drop method (deionized water). The drop shape was analyzed using the Kruss DSA30E analyzer (Hamburg, Germany). The drop volume was 4 µL. The measurements were performed for 6 droplets with an accuracy of ±5°.

#### 2.3.9. Degree of Swelling

The degree of swelling was estimated as follows: HDPE/PVA samples were allowed to stay at RH = 85% (water vapor) at room temperature until their equilibrium weight was attained. The weight of both initial and swollen samples was measured with an AND ER182A balance with an accuracy of ±0.0001 g. The equilibrium degree of swelling was read as:A = [(m_t_ − m_0_)/m_0_] × 100, % (6)
where m_t_ and m_0_ stand for the weight of the swollen and initial samples, respectively.

#### 2.3.10. Water Vapor Permeability

The vapor permeability or water vapor transmission rate (WVTR) was studied according to the Standard Test Methods ASTM E96 [21]. A cup was filled about half-way with water. The sample was placed over the top of the cup and sealed with wax on both sides so that water vapors could pass through the sample. The cup was placed in a conditioned chamber and the weighing was performed at given time intervals. The water vapor transmission rate was calculated as:K = Δm/[S × t] × 100 (%) (7)
where Δm is the weight of evaporated water at time t (days), and S is the membrane area. The measurements were performed for at least 3 samples, and the standard deviation was below 15%.

#### 2.3.11. Wide Angle X-ray Scattering (WAXS)

The WAXS analysis of nanocomposite was performed with a DIKSI station of the Kurchatov synchrotron (National Research Center Kurchatov Institute, Moscow, Russia). A 1.7 T rotating magnet operating at a radiation power of 7.65 keV (1.625 A), resolution dE/E of 10^−3^, and a photon flux of 10^9^ was used as a source of radiation. The beam size at a sample was 0.5 × 0.3 mm^2^; diffraction patterns were recorded with a Pilatus 1M detector (Dectris Ltd., Baden, Switzerland).

#### 2.3.12. Statistical Analysis

All measurements were performed at least three times for each sample. A mean value and standard deviation were estimated using Data Analysis from the Origin Pro 8 program (OriginLab Corporation, Northampton, MA, USA).

## 3. Results

### 3.1. Preparation of the HDPE/PVA Nanocomposites by Environmental Crazing

In the case of environmental crazing, the necessary condition for the development and stabilization of the fibrillar–porous structure is the presence of the PALE, which is able to reduce the surface energy and/or plasticize the polymer [17,22]. As the PALE for HDPE, water–ethanol solutions of PVA were selected, as this choice makes it possible to introduce PVA into HDPE upon tensile drawing. In this case, water serves as a solvent for PVA but not as the PALE for hydrophobic HDPE (high contact angle, low wettability, no plasticization). Hence, ethanol was selected as the second component of the PVA solution as the PALE for HDPE. In contrast to many other organic solvents, ethanol is readily mixed with water. Evidently, upon environmental crazing, the content of the guest component (PVA) in the host matrix (HDPE) depends on the PVA concentration in the feed solution. Taking into account the fact that the solubility of PVA in water decreases as the content of ethanol increases, the optimal water/ethanol composition was selected as 3/2 (vol/vol) when environmental crazing was promoted by ethanol. As a result, solutions with a PVA concentration varying from 3 to 11 wt.% were used.

Figure 1 shows the AFM images of the initial HDPE and HDPE after the deformation in the water–ethanol solution. The AFM image of the initial HDPE shows the layers of crystalline lamellae which are primarily oriented perpendicular to the extrusion axis (MD direction), and this organization is referred to as the row-nucleated structure, the Keller–Machin structure, or the layered lamellar structure [23]. After deformation of the HDPE films in the water–ethanol solutions, the AFM images show the development of numerous fibrils which bridge the neighboring crystallites, and this mode of environmental crazing is referred to as the intercrystallite crazing. The fibrils are oriented along the direction of tensile drawing and separated by the slitlike elongated pores. The environmental crazing of semicrystalline HDPE is also accompanied by the fragmentation of crystalline lamellae and by the displacement of lamellar fragments with respect to each other (Figure 1). A long period was estimated from the cross-section profiles constructed along the extrusion direction and the direction of tensile drawing. This value was ~15–30 nm for the initial HDPE and 100–250 nm for the samples stretched by 200%. The mean distance between the peaks (the parameter that characterizes the sum of the pore width and fibril diameter) was equal to ~35 nm. In other words, the environmental crazing of HDPE films promotes the development of a mesoporous structure according to the IUPAC classification [24].

The efficiency of the PALE as a crazing-promoting agent can be estimated from changes in the stress–strain behavior of polymers. Figure 2 presents the engineering stress–strain curves of HDPE upon tensile drawing in air, and in the water–ethanol solution containing PVA (the concentration of PVA in the HDPE/PVA solution was 7 wt.%). In comparison with the deformation in air, the stretching of HDPE in a PVA water–ethanol solution was accompanied by a decrease in the yield stress from 33 to 23 MPa. The stress–strain curve illustrating the tensile drawing of HDPE in the PALE (curve 2, Figure 2) is seen to be smoother and has no well-pronounced yield tooth as compared with the corresponding curve upon deformation in air.

Porosity is credited as the key parameter for all polymeric porous materials. Figure 3 shows the experimental volume porosity of the HDPE films after their tensile drawing via environmental crazing versus tensile strain ε and the theoretical curve calculated through Equation (2) under the assumption that deformation proceeds exclusively via the development of porosity (“ideal” crazing). As follows from Figure 3, at the initial stage, porosity rapidly increases with the increasing tensile strain but after ε = 100%, this growth slows down. Experimental curves (1, 2) are seen to be lower than the theoretical one (3). This deviation is provided by the coagulation processes within the porous structure due to a highly developed and thermodynamically unstable free surface (as follows from Figure 1). On passing from ethanol to the water–ethanol solution, the porosity of HDPE is seen to be lower at all tensile strains ε, and this fact is related to the lower ability of the aqueous solution to stabilize the as-formed porous structure. This tendency is most pronounced at low tensile strains when the deviation is 7 vol.%.

Therefore, according to the tensile tests, changes in volume porosity, and AFM studies, the deformation of HDPE in the water–ethanol solution containing PVA proceeds via environmental crazing and leads to the formation of a mesoporous structure, which may serve as a host matrix for the accommodation of modifying additives and the preparation of nanocomposites.

Upon environmental crazing, PVA macromolecules penetrate into the porous structure of the HDPE film, and the polymer–polymer HDPE/PVA nanocomposite is formed. This fact is supported by the gravimetric measurements. Figure 4 shows the content of PVA in the HDPE/PVA composite plotted against the tensile strain ε. As follows from Figure 4a, the content of PVA increases according to the porosity/tensile strain curve (Figure 3), up to 22 wt.% at ε = 350%. In other words, in the porous matrix with a higher porosity, the volume for the penetration of the solution is larger, and the content of PVA in the composite is higher (after the removal of the solvent); hence, the PVA content/tensile strain plot correlates with the porosity/tensile strain plot. As the concentration of PVA in the water–ethanol solution increases, the content of PVA in the HDPE/PVA composite also increases, provided all other conditions are the same (Figure 4b). The dependence of the PVA content in the composite on the concentration of PVA in the solution is linear: the higher the concentration of PVA in the solution, the higher the content of PVA in the PVA/HDPE nanocomposite.

### 3.2. Structure of the HDPE/PVA Nanocomposites

The distribution of PVA within the HDPE host matrix was studied by the EDX SEM method. Figure 5a,b shows the corresponding images of the fractured surface of the HDPE/PVA nanocomposite, and the map illustrating the distribution of oxygen (white dots) in the HDPE/PVA nanocomposites. Oxygen atoms of PVA (HDPE is oxygen-free) are seen to be uniformly distributed within the host matrix, and this evidence supports the conclusion that PVA macromolecules penetrate the porous structure of the HDPE matrix, and a uniform HDPE/PVA nanocomposite is formed. The pore walls of the HDPE film prevent both the aggregation of the incorporated polymer and the macroscopic phase separation which is observed upon blending of polymer melts or solutions.

According to the DSC analysis (Figure 5c), PVA within the HDPE matrix exists in the amorphized state: the DSC curves show the melting peak of HDPE at 128 °C (the degree of crystallinity was 63%, similar to that of the neat HDPE) and a weak step at 218 °C, which is below the melting temperature of PVA (225 °C). The degree of crystallinity of the PVA component in the HDPE/PVA composite was only 5% versus 57% upon “free” crystallization (not in pores).

Figure 6 presents the wide-angle X-ray scattering (WAXS) scan of the HDPE/PVA composite in both equatorial (1) and meridional directions (2). The X-ray scattering curves of the HDPE/PVA composite show the reflections of HDPE: 110, 200, 210, 020 at 15.4, 17.1, 21.4, and 25.7 nm^−1^. A weak 010 reflection of the monoclinic phase of HDPE at 13.9 nm^−1^ is overlapped with the most intensive reflection of PVA. The HDPE peaks are seen to be more pronounced in the equatorial direction. Hence, one may conclude that the HDPE macromolecules are oriented along the direction of tensile drawing upon environmental crazing, as shown in Figure 6 (insert). The above evidence correlates with the AFM image of the mesoporous HDPE (Figure 1), which shows the HDPE fibrils oriented along the direction of tensile drawing and crystalline lamellae in which long axes of macromolecules are also oriented along the direction of tensile drawing. Therefore, the structure of the polymer–polymer nanocomposite is governed by the structure of the mesoporous HDPE matrix.

### 3.3. Properties (Performance) of the HDPE/PVA Nanocomposites

The effect of the hydrophilic component (PVA) on the wettability of HDPE was studied. HDPE is known to be a hydrophobic polymer with a contact angle (CA) of 98–102°. The HLB of the composites is controlled by the content and hydrophilicity/hydrophobicity of their components. Figure 7 presents the CA plotted against the content of hydrophilic PVA in the HDPE/PVA composite, which depends on the tensile strain ε (Figure 4). The dependence is seen to be virtually linear, and the CA decreases when increasing the content of PVA in the HDPE/PVA composite (Figure 7). A comparison of Figure 4a and Figure 7 shows that the hydrophilization (CA) of the HDPE/PVA composite depends on the tensile strain of HDPE: as the tensile strain increases, the content of PVA in the HDPE/PVA composite increases; in turn, the higher the content of PVA in the HDPE/PVA composite, the higher the hydrophilization of HDPE. Therefore, the HLB of the composite materials prepared via environmental crazing can be controlled by the tensile strain of the host HDPE matrix. When the content of PVA in the HDPE/PVA composite is 22 wt.% (ε = 350%), the CA is equal to 45°; in other words, the incorporation of PVA leads to the hydrophilization of the HDPE matrix.

As was mentioned above, the mesoporous structure of the polymers after environmental crazing is thermodynamically unstable. A strain recovery of the samples in the free-standing state after deformation leads to a decrease in the free surface energy, and porosity is healed [25]. As follows from Figure 8, strain recovery decreases as the tensile strain ε increases due to the accumulation of irreversible structural changes (for example, a breakage and fragmentation of lamellae, shearing of lamellar fragments). The strain recovery of the nanocomposite films appears to be ~1.5–2 times lower than that of the unloaded HDPE film after environmental crazing. In other words, PVA within the pores prevents the strain recovery of the host HDPE matrix. When the samples are removed from the clamps of the stretching unit, stress relaxation takes place; as a result, the samples immediately shrink down along the direction of tensile drawing (according to the results shown in Figure 8 where the strain recovery (shrinkage) is plotted against the tensile strain of HDPE). The initial fast stage of the strain recovery was followed by a slow stage, and, within the following 3 days, the strain recovery was only 1–3%, which means that the equilibrium strain recovery was achieved, and the sample preserved its dimensions (this fact was proved by observations over 1 year). The composition of the nanocomposite materials remained unchanged.

The structure of the HDPE/PVA nanocomposite can be presented as a network of channels in the hydrophobic host matrix (HDPE) loaded with the hydrophilic component (PVA). Despite the strain recovery, the HDPE/PVA retained its ability to swell in water, and this fact indicates that the sample contained residual “unhealed” porosity. The HDPE/PVA nanocomposite was allowed to stay in a humid atmosphere at the relative humidity RH = 85%; the net degree of swelling was 4 wt.%. Taking into account the fact that HDPE is unable to sorb water, the degree of swelling normalized by the PVA content was 27 wt.%. Hence, when one side of the HDPE/PVA film contacted the water vapors, water uptake took place, but this water evaporated from the other side of the film where the relative humidity was lower. In this manner, water vapors permeated through the film, and the HDPE/PVA composite showed the performance of a breathable vapor-permeable material. The vapor permeability or water vapor transmission rate (WVTR) of the HDPE/PVA materials depended on the content of PVA, and this value varied from 226 to 856 g/(m^2^ day) for the nanocomposites based on the host HDPE matrixes with ε = 100% and 200%, respectively.

## 4. Discussion

The tensile drawing of HDPE films in water–ethanol solutions containing biocompatible PVA proceeds via environmental intercrystallite crazing and promotes the development of the fibrillar–mesoporous structure of HDPE (Figure 1), which serves as the host matrix for PVA.

The intercrystallite crazing (IC) of semicrystalline polymers is fundamentally different from the classical crazing (CC) of amorphous glassy polymers [15,16,17]: in the morphology and size of crazes (the walls of meso-sized IC crazes are crystallites; in CC, micron-sized crazes lie between sections of undeformed polymer) and in the mechanism of nucleation (in CC, crazes nucleate on surface defects; in IC, the leading role belongs to cavitations arising in the amorphous phase, which develop in pores under a supporting stress with the participation of PALE). In comparison with the deformation in air, the stretching of HDPE in a PVA water–ethanol solution is accompanied by a decrease in the yield stress from 33 to 23 MPa (Figure 2), which is typical for adsorption-active liquids that can reduce the surface energy and facilitate the formation of a new surface—the Rehbinder effect in polymers [22].

The environmental crazing of HDPE films is accompanied by the development of a mesoporous structure, and a polymer solution is hovered inside the film under the action of the negative hydrostatic pressure. According to the scaling concept [26], macromolecules are able to penetrate into the nanopores via segmental reptation movements. The principal condition for the penetration of macromolecules into nanopores from dilute solutions is the following rule, R_h_ ≤ D, where D is the pore diameter, and R_h_ is the hydrodynamic coil radius; in semidilute solutions, ξ ≤ D, where ξ is the blob size (correlation length as the distance between the neighboring joints of the entanglement network). For PVA with 150 kD, the hydrodynamic radius of the coil is calculated as R_h_ = 0.0202 (M_w_^0.58^) [27,28], and this value is ~20 nm. Hence, macromolecules can easily penetrate the pores (according to the AFM observations, ~35 nm) of the porous HDPE even from the dilute solutions. Hence, PVA together with the solution occupies the pores of the mesoporous HDPE matrix, and the polymer–polymer HDPE/PVA nanocomposite is formed after solvent removal. Crazing allows the introduction of up to 22 wt.% of PVA into the HDPE matrix (Figure 4a). Upon tensile drawing, the solution of PVA uniformly occupies the porous HDPE matrix, and both pore walls and fibrils prevent the aggregation of PVA (Figure 5a,b).

Nanoscale dimensions of the fibrillar–porous HDPE create the conditions for the confined crystallization of PVA, which exists in the amorphized state according to the DSC data (Figure 5c). This evidence agrees with the literature data [29,30] and can be explained by the spatial confinements of the mesoporous polymer matrix when the crystallization of the second polymer component (polyethylene oxide) within the pores is hindered or fully prevented [31]. The tensile drawing of HDPE along the extrusion direction upon environmental crazing leads to the orientation of macromolecules along the direction of stretching (Figure 6).

The tensile strain of the HDPE host matrix and the concentration of PVA in the water–ethanol solutions make it possible to control the content of PVA in the HDPE/PVA nanocomposite (Figure 4) and the surface area occupied by the hydrophilic PVA. In this case, the wettability or HLB of the HDPE/PVA nanocomposites can be varied in a broad interval from 98 to 45° (Figure 7). The well-known methods for the hydrophilization of polyolefins are the following: plasma treatment, ozonation, graft polymerization, and the incorporation of hydrophilic additives. However, the above methods require special equipment, and in many cases, they fail to achieve a high wettability. Hence, environmental crazing can be considered as an efficient route for the hydrophilization of the hydrophobic HDPE via the introduction of a hydrophilic component and the preparation of the polymer–polymer nanocomposite.

Evidently, PVA within the pores of the HDPE host matrix prevents the strain recovery of the composite material. However, despite the strain recovery, the HDPE/PVA retains its ability to swell in water (4 wt.%), and this fact indicates that the sample contains residual “unhealed” porosity. The ability of HDPE/PVA samples to sorb water allows their use as breathable materials with vapor permeability up to 856 g/(m^2^ day).

The modification of HDPE by biocompatible PVA allows one to broaden the scope of practical applications of commercial HDPE as biomedical, packaging, and protective materials, modern textile articles, breathable materials, separation membranes, selective sorbents, etc. This approach can be implemented using traditional equipment for the orientational drawing of polymers, and hydrophilization via environmental crazing can be considered as processing and secondary use of polyolefins.

## Figures and Tables

**Figure 1 polymers-16-01184-f001:**
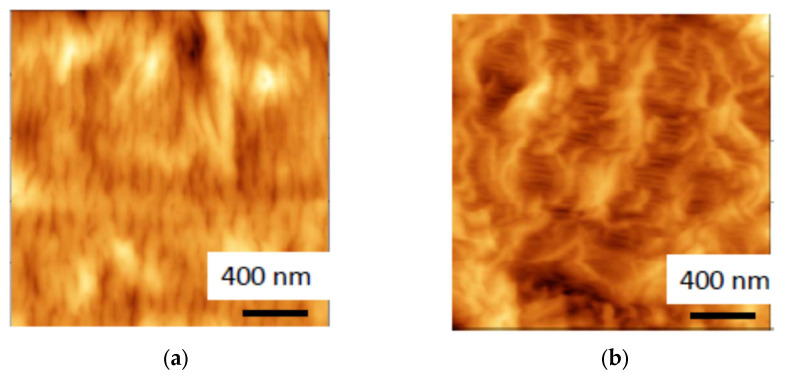
The AFM images of (**a**) initial HDPE film and (**b**) HDPE film after tensile drawing by 200% via environmental crazing, the extrusion direction and direction of tensile drawing are horizontal.

**Figure 2 polymers-16-01184-f002:**
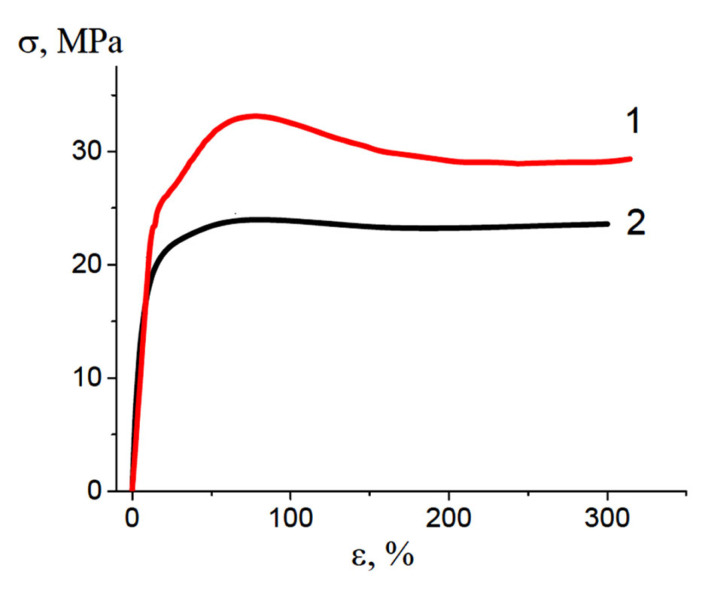
Stress–strain curves (initial portion) illustrating tensile drawing of HDPE along the MD direction in air (1) and in the water–ethanol solution containing PVA (2).

**Figure 3 polymers-16-01184-f003:**
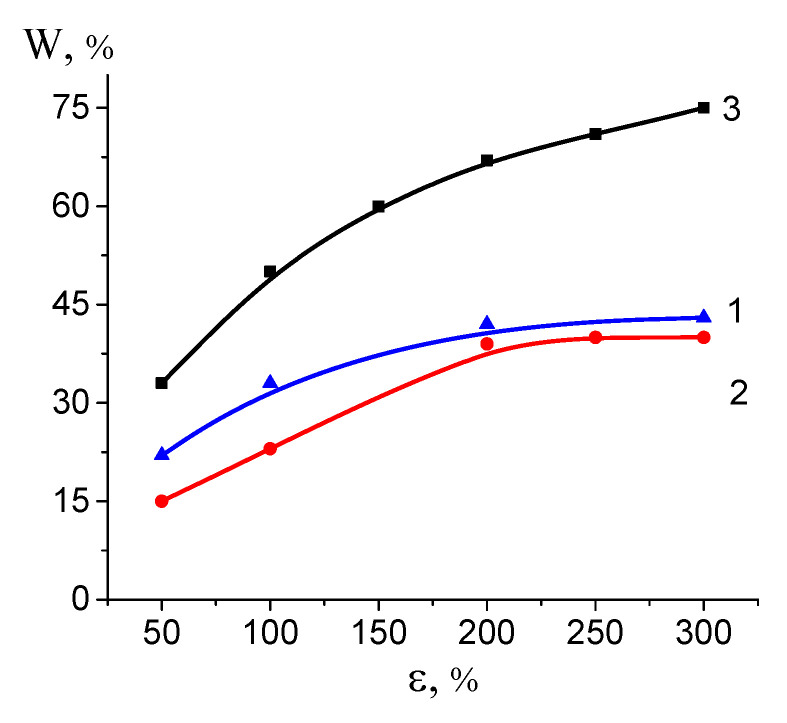
Volume porosity W versus tensile strain ε upon tensile drawing of HDPE in (1) ethanol, (2) water–ethanol solution, and (3) theoretical dependence for an “ideal” crazing.

**Figure 4 polymers-16-01184-f004:**
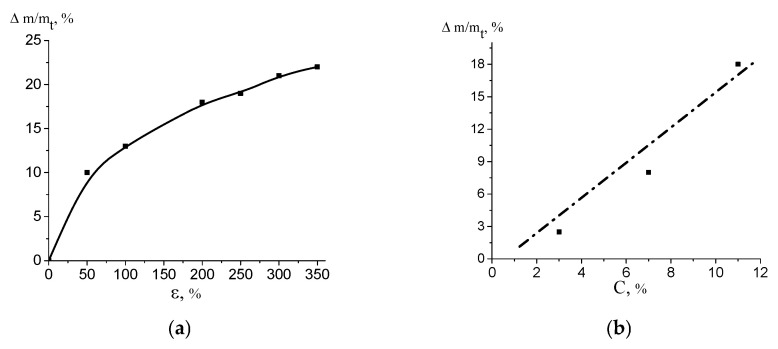
(**a**) Content of PVA in the HDPE/PVA nanocomposite plotted against tensile strain ε (the concentration of PVA in the water–ethanol solution was 11 wt.%); (**b**) content of PVA in the HDPE/PVA nanocomposite plotted against the concentration of PVA in the water–ethanol solution (HDPE ε = 200%).

**Figure 5 polymers-16-01184-f005:**
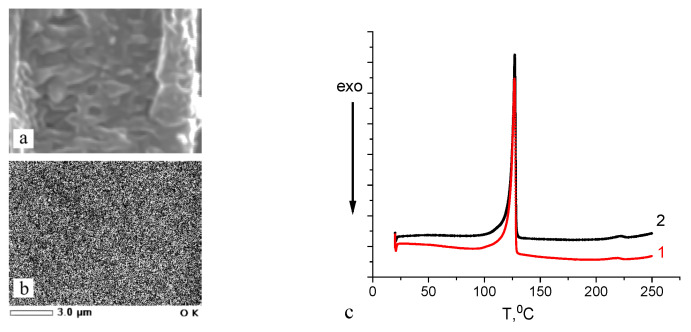
(**a**) SEM micrograph of the fractured surface of the HDPE/PVA nanocomposite (the PVA content was 18 wt.%); (**b**) the map illustrating the distribution of oxygen (white dots) in the HDPE/PVA nanocomposites; (**c**) DSC heating scans of the HDPE/PVA nanocomposite based on HDPE with (1) ε = 200% and (2) ε = 300%; the heating/cooling rate was 10 K/min.

**Figure 6 polymers-16-01184-f006:**
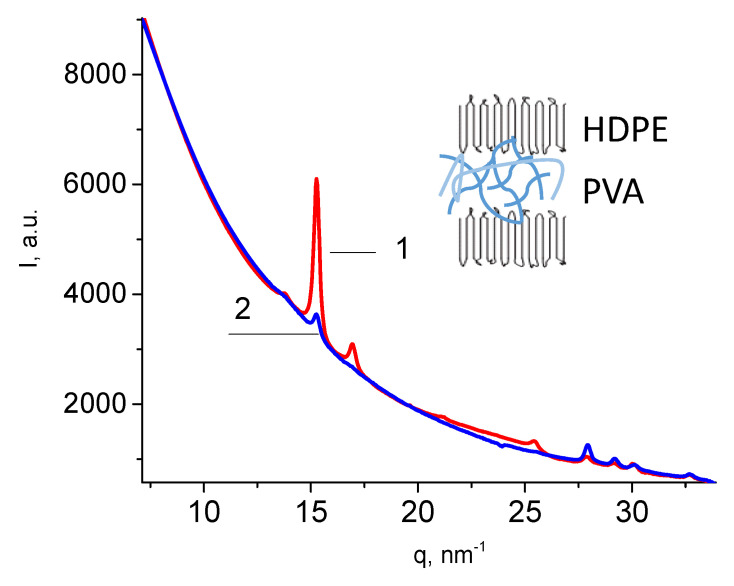
Diffractograms of the HDPE/PVA composite in the equatorial (1) and meridional directions (2); the scheme illustrating the structure of the HDPE/PVA composite is shown in the insert.

**Figure 7 polymers-16-01184-f007:**
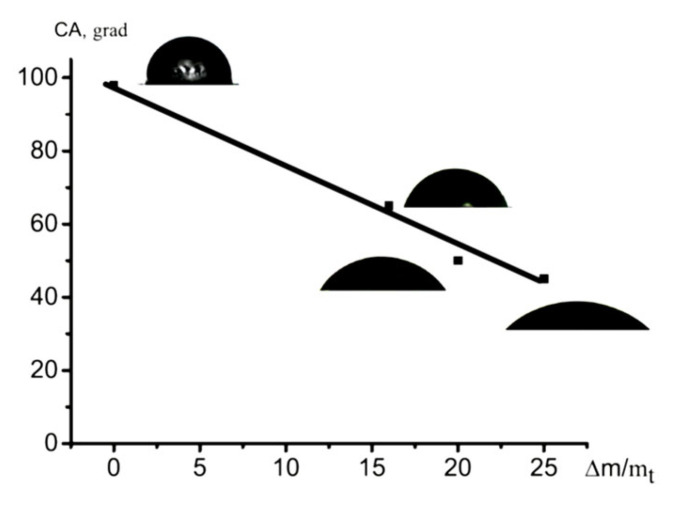
Contact angle against the content of PVA in the HDPE/PVA composites.

**Figure 8 polymers-16-01184-f008:**
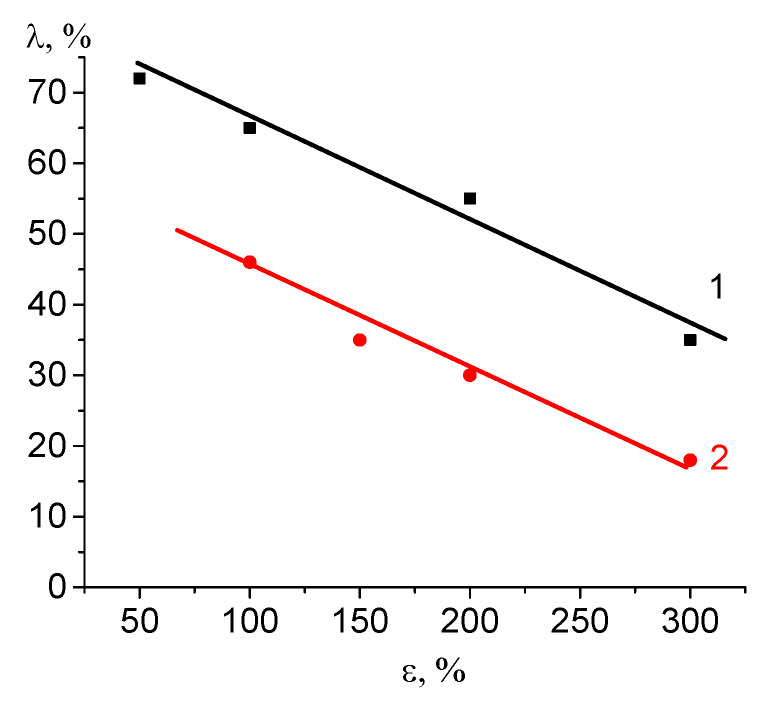
Strain recovery under stress relaxation versus tensile strain for (1) the HDPE films after tensile drawing to different tensile strains via environmental crazing and (2) HDPE/PVA composite.

## Data Availability

Data are contained within the article.

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
