# Peer review of "Modification of High-Density Polyethylene with a Fibrillar–Porous Structure by Biocompatible Polyvinyl Alcohol via Environmental Crazing"

_polymers, 2024, doi:10.3390/polym16091184_

Round 1

Reviewer 1 Report

Comments and Suggestions for Authors

A positive aspect of this manuscript is the exploration of a method for creating HDPE/PVA nanocomposites via environmental crazing. This approach offers a way to modify the properties of HDPE by incorporating biocompatible PVA, potentially opening up new avenues for various applications such as biomedical materials, packaging, textiles, and membranes.

A non clear point of this manuscript could be the potential challenges associated with the stability of the mesoporous structure formed during the environmental crazing process. While the method offers an approach for creating HDPE/PVA nanocomposites, the thermodynamic instability of the porous structure might pose difficulties in maintaining the desired properties over time, especially in practical applications where long-term stability is crucial.
There are some questions, suggestions for modifications or improvements:

- The introduction could benefit from a clearer delineation of the problem statement, objectives, and relevance of the study.
- How does the environmental crazing process differentiate itself from conventional methods of polymer modification?
- What are the specific challenges associated with maintaining the stability and performance of the HDPE/PVA nanocomposites, particularly in applications where long-term durability is essential?

Line 248: "Degree of crystallinity of the PVA component in the HDPE/PVA composite is only 5% versus 57% upon «free» crystallization." It's not clear what "free crystallization" refers to and how it differs from the crystallinity observed in the composite.
Line 255: "Hence, one may conclude that the HDPE macromolecules are oriented along the direction of tensile drawing upon environmental crazing, as shown in Fig. 6 (insert)." The significance of the orientation of HDPE macromolecules along the direction of tensile drawing and its relation to environmental crazing needs further clarification.
Line 269: “Therefore, HLB of the composite materials prepared via environmental crazing can be controlled by the tensile strain of the host HDPE matrix."  It's unclear how tensile strain controls the hydrophilic-lipophilic balance (HLB) of composite materials and what implications this has for their properties.
Line 293: “The HDPE/PVA samples are able to sorb water, and this ability allows their use as breathable materials with vapor permeability." While it's understood that water sorption affects the breathability of materials, the mechanism by which HDPE/PVA samples sorb water and its relationship to vapor permeability could be explained more explicitly.
Line 313: "Upon crazing, PVA macromolecules penetrate into the porous structure of the HDPE film, and the polymer-polymer HDPE/PVA nanocomposite is formed." The process of PVA macromolecule penetration into the HDPE film's porous structure and its role in forming the nanocomposite could be described in more detail.

Author Response

Reviewer#1

A positive aspect of this manuscript is the exploration of a method for creating HDPE/PVA nanocomposites via environmental crazing. This approach offers a way to modify the properties of HDPE by incorporating biocompatible PVA, potentially opening up new avenues for various applications such as biomedical materials, packaging, textiles, and membranes.

A non clear point of this manuscript could be the potential challenges associated with the stability of the mesoporous structure formed during the environmental crazing process. While the method offers an approach for creating HDPE/PVA nanocomposites, the thermodynamic instability of the porous structure might pose difficulties in maintaining the desired properties over time, especially in practical applications where long-term stability is crucial.

Answer

We are very thankful for the valuable comments which allowed us to improve the quality of our manuscript. All changes and revisions are marked by the yellow color.

When the samples are removed from the clamps of the stretching unit, stress relaxation takes place; as a result, the samples immediately shrink down along the direction of tensile drawing (according to the results shown in Fig. 8 where the strain recovery (shrinkage) is plotted against the tensile strain of HDPE). Strain recovery decreases as the tensile strain increases, and this fact is explained by the increased contribution from irreversible modes of deformation and irreversible changes in the polymer structure. The process of strain recovery proceeds with time. The initial fast stage of strain recovery (18-42 % for HDPE/PVA and 32-70%  for HDPE) is followed by the slow stage, and, within the following 3 days, the strain recovery is only 1-3%. In other words, equilibrium strain recovery is attained within 3 days. After 3 days, shrinkage is completed, and the samples do not change their geometrical dimensions (this fact is proved by our observations within 1 year). Composition of the nanocomposite materials remains unchanged because the solvent is removed before the strain recovery (in vacuum under isometric conditions as described in Experimental Part), and PVA is accommodated within the pores of the HDPE host matrix. Therefore, environmental crazing allows preparation of the stable HDPE-PVA nanocomposites.

The text of the manuscript is revised according to this comment.

There are some questions, suggestions for modifications or improvements:

- The introduction could benefit from a clearer delineation of the problem statement, objectives, and relevance of the study.

Answer

Introduction was revised and now this section includes the statement of the problem, objectives, and relevance of the study. All changes are marked.

- How does the environmental crazing process differentiate itself from conventional methods of polymer modification?

Answer

Рreparation of sustainable multicomponent polymeric materials presents a challenging scientific and technological task, especially, when in-compatible components with different physicochemical characteristics (hydrophilici-ty/hydrophobicity, melting temperature, the Hildebrand solubility parameter, etc.) are involved. Polymer-polymer blends can be prepared by different methods from blending of polymeric solutions or melts and coextrusion [7-9] up to the development of inter-penetrating networks via one-stage or two-stage polymerization [10,11]. Noteworthy is that all methods have their benefits and limitations, for example, blending of PVA and polyolefins requires the use of plasticizing agents which markedly reduce the melting temperature of PVA [3,12,13]. Phase separation and aggregation of the components can be prevented by the addition of compatibilizers (maleic anhydride) [14].

In contrast to the above methods and approaches, environmental crazing allows the development of the mesoporous polymer matrix in one stage at room temperatures and preparation of the nanocomposite via the introduction of PVA. In this case, preparation of the multicomponent system containing a hydrophobic (HDPE) and a hydrophilic polymer (PVA) doesn’t require the use of any compatibilizers, special reagents, or complex equipment (for example, extruders). Tensile drawing of the HDPE films can be performed using a typical industrial equipment for the orientational drawing of polymers (with its minor modification).

- What are the specific challenges associated with maintaining the stability and performance of the HDPE/PVA nanocomposites, particularly in applications where long-term durability is essential?

Answer

In the case under study, no problems related to the long-term stability of the nanocomposite HDPE-PVA materials are detected. Strain recovery proceeds quickly and completely once the stress is relieved (upon stress relaxation). After 3 days, the sample preserve their shape stability, and their geometrical dimensions remain unchanged.

Line 248:"Degree of crystallinity of the PVA component in the HDPE/PVA composite is only 5% versus 57% upon «free» crystallization." It's not clear what "free crystallization" refers to and how it differs from the crystallinity observed in the composite.

Answer

In this work, “free crystallization” means bulk crystallization under conventional conditions (in contrast to the confined crystallization within nanoscale space of pores). This definition is added to the manuscript.

Line 255: "Hence, one may conclude that the HDPE macromolecules are oriented along the direction of tensile drawing upon environmental crazing, as shown in Fig. 6 (insert)." The significance of the orientation of HDPE macromolecules along the direction of tensile drawing and its relation to environmental crazing needs further clarification.

Answer

Orientation of HDPE macromolecules upon environmental crazing presents an evident scientific interest as the manifestation of plastic deformation of semicrystalline polymers due to their deformation in the presence of physically active liquid environments. The corresponding AFM image (Fig. 1) shows the orientation of the polymeric material as fibrils directed along the direction of tensile drawing. At the molecular level, this fact is also confirmed by the X-ray analysis. This evidence is important for the correct understanding of processes and structural transformations upon environmental crazing.

Line 269:“Therefore, HLB of the composite materials prepared via environmental crazing can be controlled by the tensile strain of the host HDPE matrix."  It's unclear how tensile strain controls the hydrophilic-lipophilic balance (HLB) of composite materials and what implications this has for their properties.

Answer

Figure 7 shows the contact angle (CA, wettability) plotted against the content of PVA in the HDPE-PVA nanocomposite. As follows from Fig. 7, the decrease in CA is associated with an increase in the PVA content. Figure 4a shows that the content of PVA in the HDPE-PVA nanocomposite depends on the tensile strain. Hence, with varying the tensile strain, content of PVA and wettability of the HDPE-PVA nanocomposite are changed. Therefore, tensile strain is the key factor that controls the wettability (HLB) of the system. The text of the manuscript is revised.

Line 293: “The HDPE/PVA samples are able to sorb water, and this ability allows their use as breathable materials with vapor permeability." While it's understood that water sorption affects the breathability of materials, the mechanism by which HDPE/PVA samples sorb water and its relationship to vapor permeability could be explained more explicitly.

Answer

In the revised version of the manuscript, the results on swelling and vapor permeability are presented in the same section. The results on vapor permeability are described in more detail. The channels in the mesoporous structure of the HDPE host matrix are loaded with the hydrophilic PVA which is able to sorb water not only upon the direct content with water but also with water vapors. Hence, when one side of the HDPE-PVA film contacts the water vapors, water uptake takes place, but this water evaporates from the other side of the film where the relative humidity is lower. In this manner, water vapors permeate through the film, and the HDPE-PVA composite shows the performance of a breathable vapor-permeable material.

Line 313: "Upon crazing, PVA macromolecules penetrate into the porous structure of the HDPE film, and the polymer-polymer HDPE/PVA nanocomposite is formed." The process of PVA macromolecule penetration into the HDPE film's porous structure and its role in forming the nanocomposite could be described in more detail.

Environmental crazing of HDPE films is accompanied by the development of a mesoporous structure, and a polymer solution is hovered inside the film under the action of the negative hydrostatic pressure.  According to the scaling concept [28], macromolecules are able to penetrate into the nanopores via segmental reputational movements. The principal condition for the penetration of macromolecules into na-nopores from dilute solutions is the following rule Rh ≤ D, where D is the pore diameter, Rh is the hydrodynamic coil radius; in semidilute solutions ξ ≤ D, where ξ is the blob size (correlation length as the distance between the neighboring joints of the entanglement network). For PVA with 150 kD, hydrodynamic radius of the coil is calculated as Rh = 0.0202 (Mw0.58) [29, 30], and this value is ~20 nm. Hence, macromolecules can easily penetrate the pores (according to the AFM observations, ~35 nm) of the porous HDPE even from the dilute solutions. Hence, PVA solution penetrates the pores of the mesoporous HDPE matrix, and the polymer-polymer HDPE/PVA nanocomposite is formed after solvent removal. Crazing allows introduction of up to 22 wt.% of PVA to HDPE matrix (Fig. 4a). Upon tensile drawing the solution of PVA uniformly occupies the porous HDPE matrix, and both pore walls and fibrils prevent the aggregation of PVA (Fig. 5a,b). Preparation of the HDPE-PVA nanocomposites is described in more detail in the Experimental section of the manuscript. The manuscript is revised and new information concerning the penetration of PVA into the HDPE host matrix is added.

Reviewer 2 Report

Comments and Suggestions for Authors

The current work has a certain merit but is still more a report than a dedicated scientific contribution. I detected no extremely major flaws but the writing should be somewhat polished, as explained below.

Introduction: Molecular weight should be better molar mass

Results and Discussion:

Pleas highlight more the structure expected; also make a better link to the novelty by comparing with the leading property values in the field but also mentioning the upgrades.

A general reader needs more details on the procedure to obtain the porosity values.

Figure  needs more discussion.

Author Response

Reviewer #2

The current work has a certain merit but is still more a report than a dedicated scientific contribution. I detected no extremely major flaws but the writing should be somewhat polished, as explained below.

Answer

We are grateful for the valuable comments which allowed us to improve the quality of our manuscript. All changes and revisions are marked by the yellow color.

Introduction: Molecular weight should be better molar mass

Answer

We agree. The term «molecular weight» is used in the text.

Results and Discussion:

Pleas highlight more the structure expected; also make a better link to the novelty by comparing with the leading property values in the field but also mentioning the upgrades.

Answer

The key result of this scientific work is concerned with the fact that environmental crazing offers a unique route for the preparation of stable polymer-polymer nanocomposites based on polymers with different nature – hydrophilic PVA and hydrophobic HDPE. Introduction of PVA into the HDPE matrixes allows hydrophilization of the resultant material when the contact angle decreases from 98° (initial HDPE) down to 45°. In Introduction, various methods and approaches for the preparation of polymer-polymer blends and composites are revisited. However, most the proposed methods involve laborious and multistage processes, and require a special equipment. In some cases, they fail to produce the materials with the desired shape stability. When considering the methods for the hydrophilization of polymers (polyolefins), one should mention the most efficient approaches such as: plasma treatment, ozonation, graft polymerization, and incorporation of hydrophilic additives. However, the best achievements are the following: CA of HDPE decreases only down to 60-70°. Environmental crazing can be performed on the industrial equipment for the orientational drawing of polymers (with minor modifications). Hence, as compared with other methods, environmental crazing offers an efficient and inexpensive approach for the modification of polymers and preparation of new polymeric materials.

The text of the manuscript is revised.

A general reader needs more details on the procedure to obtain the porosity values.

The text is revised, and the protocol for the estimation of porosity is described in more detail:

To study the mechanism of plastic deformation of HDPE, volume porosity (as the fractional content of pores) is estimated as  

W = [ΔV / Vt] × 100 (%) (1)

where ΔV is the volume gain, and Vt is the current volume of the sample at a given tensile strain ε. Volume of the samples was estimated from the geometrical dimensions of the test samples (length. thickness, width) using an IZV2 optimeter; experimental error was below ± 0.5μm (thickness).

Figure  needs more discussion.

The description of the figures was added.

Reviewer 3 Report

Comments and Suggestions for Authors

The manuscript submitted by Yarysheva  and her colab. describes the preparation and structural  characterization of nanocomposites based on HDPE matrix and PVA as a host. The nanocomposites were obtained by tensile drawing of HDPE films in mixed solution.

The subject is interesting from the theoretical point of view, being included in the physical chemistry of polymers, and from a practical point of view, as suggesting a new type of composite polymeric material for different applications.

The experimental part is presented well and sufficiently detailed, and the analytical methods are those usually utilized for morphological characterization.

I do have several issues, as follow:

1. The nanocomposite analyses are focused on morphology, because is correlated with the practical performances. I am sure the authors know that the vibrational spectroscopy is very informative about the (micro)structure of both HDPE matrix and especially of PVA. The amorphous state of PVA has been indicated by DSC only, which is fine, but it would have been observed very easily in a FTIR analysis. And the junction points between HDPE and PVA would be visible as well.

2. If the nancomposite is thermodinamically unstable and the structural changes that appear modify the microporosity, what are the long term effects in case of real applications? It is stated that it can be used as breathable material, but it is also indicated that the porosity is healed.

Apart of these two little problems, the manuscript is well written and well presented. The Figures are clear, the analyses are correct interpreted and the conclusions sound justified.

I consider the manuscript does not carry serious issues and it can be published after the authors will answer to the two questions.

Author Response

Reviewer #3

The manuscript submitted by Yarysheva  and her colab. describes the preparation and structural  characterization of nanocomposites based on HDPE matrix and PVA as a host. The nanocomposites were obtained by tensile drawing of HDPE films in mixed solution.

The subject is interesting from the theoretical point of view, being included in the physical chemistry of polymers, and from a practical point of view, as suggesting a new type of composite polymeric material for different applications.

The experimental part is presented well and sufficiently detailed, and the analytical methods are those usually utilized for morphological characterization.

I do have several issues, as follow:

  1. The nanocomposite analyses are focused on morphology, because is correlated with the practical performances. I am sure the authors know that the vibrational spectroscopy is very informative about the (micro)structure of both HDPE matrix and especially of PVA. The amorphous state of PVA has been indicated by DSC only, which is fine, but it would have been observed very easily in a FTIR analysis. And the junction points between HDPE and PVA would be visible as well.

Answer

We are very thankful for the valuable comments which allowed us to improve the quality of our manuscript. All changes and revisions are marked by the yellow color.

We agree. According to the following publication [O. N. Tretinnikov, S. A. Zagorskaya Determination of the degree of crystallinity of poly(vinylalcohol) by FTIR spectroscopy, Journal of Applied Spectroscopy, Vol. 79, No. 4, 2012] the degree of crystallinity of PVA can be estimated from the peak at 1144 cm–1 and the ratio of intensities of the peaks 1144/1094 cm–1. The analysis of the HDPE-PVA composites was performed. We found not only the state of PVA, but also an evident difference in the intensities of the peaks for the samples along and perpendicular to the direction of tensile drawing (dichroism, indicating the orientation of macromolecules). Unfortunately, this work presents an independent direction of our studies and now we are unable to use these results in this manuscript.

  1. If the nancomposite is thermodynamically unstable and the structural changes that appear modify the microporosity, what are the long term effects in case of real applications? It is stated that it can be used as breathable material, but it is also indicated that the porosity is healed.

Answer

When the samples are removed from the clamps of the stretching unit, stress relaxation takes place; as a result, the samples immediately shrink down along the direction of tensile drawing (according to the results shown in Fig. 8 where the strain recovery (shrinkage) is plotted against the tensile strain of HDPE). Strain recovery decreases as the tensile strain increases, and this fact is explained by the increased contribution from irreversible modes of deformation and irreversible changes in the polymer structure. The process of strain recovery proceeds with time. The initial fast stage of strain recovery (18-42 % for HDPE/PVA and 32-70%  for HDPE)  is followed by the slow stage, and, within the following 3 days, the strain recovery is only 1-3%. In other words, equilibrium strain recovery is attained within 3 days. After the 3 days, shrinkage is completed, and the samples do not change their geometrical dimensions (this fact is proved by the observations within 1 year). Composition of the nanocomposite materials remains unchanged because the solvent is removed before the strain recovery (in vacuum under isometric conditions as described in Experimental Part), and PVA is accommodated within the pores of the HDPE host matrix. Therefore, environmental crazing allows preparation of stable HDPE-PVA nanocomposites.

The text of the manuscript is revised according to this comment.

As follows from Fig. 8, strain recovery of the HDPE-PVA (curve 2) is lower than that of the HDPE host matrix (curve 1) at all tensile strains, and this fact can be explained by the presence of PVA within the pores of the HDPE matrix. Despite the strain recovery, the polymer matrix preserves the residual porosity, and PVA in the HDPE-PVA composite is able to swell; in other words, the PVA phase serves as the channels for the transmission of water vapors upon sorption/ desorption from both sides of the composite membrane.

Hydrophilization of HDPE by PVA allows one to broaden the scope of practical applications of commercial HDPE as biomedical, packaging, and protective materials, modern textile articles, breathable materials, separation membranes, selective sorbents, etc.

Apart of these two little problems, the manuscript is well written and well presented. The Figures are clear, the analyses are correct interpreted and the conclusions sound justified.

I consider the manuscript does not carry serious issues and it can be published after the authors will answer to the two questions.

Round 2

Reviewer 1 Report

Comments and Suggestions for Authors

 Accept in present form